Host-Microbe Biology
# Bactopia: a Flexible Pipeline for Complete Analysis of Bacterial Genomes

Robert A. Petit III,ᵃ Timothy D. Readᵃ

ᵃDivision of Infectious Diseases, Department of Medicine, Emory University School of Medicine, Atlanta, Georgia, USA

**ABSTRACT** Sequencing of bacterial genomes using Illumina technology has become such a standard procedure that often data are generated faster than can be conveniently analyzed. We created a new series of pipelines called Bactopia, built using Nextflow workflow software, to provide efficient comparative genomic analyses for bacterial species or genera. Bactopia consists of a data set setup step (Bactopia Data Sets [BaDs]), which creates a series of customizable data sets for the species of interest, the Bactopia Analysis Pipeline (BaAP), which performs quality control, genome assembly, and several other functions based on the available data sets and outputs the processed data to a structured directory format, and a series of Bactopia Tools (BaTs) that perform specific postprocessing on some or all of the processed data. BaTs include pan-genome analysis, computing average nucleotide identity between samples, extracting and profiling the 16S genes, and taxonomic classification using highly conserved genes. It is expected that the number of BaTs will increase to fill specific applications in the future. As a demonstration, we performed an analysis of 1,664 public *Lactobacillus* genomes, focusing on *Lactobacillus crispatus*, a species that is a common part of the human vaginal microbiome. Bactopia is an open source system that can scale from projects as small as one bacterial genome to ones including thousands of genomes and that allows for great flexibility in choosing comparison data sets and options for downstream analysis. Bactopia code can be accessed at https://www.github.com/bactopia/bactopia.

**IMPORTANCE** It is now relatively easy to obtain a high-quality draft genome sequence of a bacterium, but bioinformatic analysis requires organization and optimization of multiple open source software tools. We present Bactopia, a pipeline for bacterial genome analysis, as an option for processing bacterial genome data. Bactopia also automates downloading of data from multiple public sources and species-specific customization. Because the pipeline is written in the Nextflow language, analyses can be scaled from individual genomes on a local computer to thousands of genomes using cloud resources. As a usage example, we processed 1,664 *Lactobacillus* genomes from public sources and used comparative analysis workflows (Bactopia Tools) to identify and analyze members of the *L. crispatus* species.

**KEYWORDS** annotation, assembly, bacteria, genomics, *Lactobacillus*, software

Sequencing a bacterial genome, an activity that once required the infrastructure of a dedicated genome center, is now a routine task that even a small laboratory can undertake. Many open-source software tools have been created to handle various parts of the process of using raw read data for functions such as single nucleotide polymorphism (SNP) calling and *de novo* assembly. As a result of dedicated community efforts, it has recently become much easier to locally install these bioinformatic tools through package managers (Bioconda [1] and Brew [2]) or through the use of software containers (Docker and Singularity). Despite these advances, producers of bacterial sequence data face a bewildering array of choices when considering how to perform

Address correspondence to Timothy D. Read, tread@emory.edu.

We have created Bactopia, a nextflow pipeline, for processing bacterial genomes and tested it on the Lactobacillus genus.

analysis, particularly when large numbers of genomes are involved and processing efficiency and scalability become major factors.

Efficient bacterial multigenome analysis has been hampered by three missing functionalities. First is the need to have workflows of workflows' that can integrate analyses and provide a simplified way to start with a collection of raw genome data, remove low-quality sequences, and perform the basic analytic steps of de novo assembly, mapping to reference sequence, and taxonomic assignment. Second is the desire to incorporate user-specific knowledge of the species into the input of the main genome analysis pipeline. While many microbiologists are not expert bioinformaticians, they are experts in the organisms they study. Third is the need to create an output format from the main pipeline that could be used for future customized downstream analysis such as pan-genome analysis and basic visualization of phylogenies.

Here, we introduce Bactopia, an integrated suite of workflows primarily designed for flexible analysis of Illumina genome sequencing projects of bacteria from the same taxon. Bactopia is based on Nextflow workflow software (3) and is designed to be scalable, allowing projects as small as a single genome to be run on a local desktop or projects including many thousands of genomes to be run as a batch on a cloud infrastructure. Running multiple tasks on a single platform standardizes the underlying data quality used for gene and variant calling between projects run in different laboratories. This structure also simplifies the user experience. In Bactopia, complex multigenome analysis can be run in a small number of commands. However, there are myriad options for fine-tuning data sets used for analysis and the functions of the system. The underlying Nextflow structure ensures reproducibility. To illustrate the functionality of the system, we performed a Bactopia analysis of 1,664 public genome samples of the *Lactobacillus* genus, an important component of the microbiome of humans and animals.

## RESULTS

**Design and implementation.** Bactopia links together open-source bioinformatics software, available from Bioconda (1), using Nextflow (3). Nextflow was chosen for its flexibility: Bactopia can be run locally, on clusters, or on cloud platforms with simple parameter changes. It also manages the parallel execution of tasks and creates checkpoints allowing users to resume jobs. Nextflow automates installation of the component software of the workflow through integration with Bioconda. For ease of deployment, Bactopia can be installed either through Bioconda, a Docker container, or a Singularity container. All of the software programs used by Bactopia (version 1.4.0) described in the manuscript are listed in Table 1 with their individual version numbers.

There are three main components of Bactopia (Fig. 1; see also Fig. S1 in the supplemental material). Bactopia Data Sets (BaDs) is a framework for formatting organism-specific data sets to be used by the downstream analysis pipeline. The Bactopia Analysis Pipeline (BaAP) is a customizable workflow for the analysis of individual bacterial genome projects that is an extension and generalization of the previously published *Staphylococcus aureus*-specific Staphopia Analysis Pipeline (StAP) (4). The inputs to BaAP are FASTQ files from bacterial Illumina sequencing projects, either imported from the National Centers for Biotechnology Information (NCBI) Short Read Archive (SRA) database or provided locally, and any reference data in the BaDs. Bactopia Tools (BaTs) is a set of workflows that use the output files from a BaAP project to run genomic analysis on multiple genomes. For this project we used BaTs to (i) summarize the results of running multiple bacterial genomes through BaAP, (ii) extract 16S gene sequences and create a phylogeny, (iii) assign taxonomic classifications with the Genome Taxonomy Database (GTDB) (5), (iv) determine subsets of *Lactobacillus crispatus* samples by average nucleotide identity (ANI) with FastANI (6), and (v) run pan-genome analysis for *L. crispatus* using Roary (7) and create a core-genome phylogeny.

**Comparison to similar open-source software.** At the time of writing (February 2020), we knew of only three other actively maintained open-source generalist bacterial

**TABLE 1** List of bioinformatic tools used by the Bactopia Analysis Pipeline, version 1.4.0

| Name | Version | Description[a] | Link | Reference(s) |
|---|---|---|---|---|
| AMRFinder+ | 3.6.7 | Finds acquired antimicrobial resistance genes and some point mutations in protein or assembled nucleotide sequences | https://github.com/ncbi/amr | 47 |
| Aragorn | 1.2.38 | Finds transfer RNA (tRNA) features | http://130.235.244.92/ARAGORN/Downloads/ | 85 |
| Ariba | 2.14.4 | Antimicrobial resistance identification by assembly | https://github.com/sanger-pathogens/ariba | 13 |
| ART | 2016.06.05 | A set of simulation tools to generate synthetic next-generation sequencing reads | https://www.niehs.nih.gov/research/resources/software/biostatistics/art/index.cfm | 59 |
| assembly-scan | 0.3.0 | Generates basic stats for an assembly | https://github.com/rpetit3/assembly-scan | 73 |
| Barrnap | 0.9 | Bacterial ribosomal RNA predictor | https://github.com/tseemann/barrnap | 86 |
| BBMap | 38.76 | A suite of fast, multithreaded bioinformatics tools designed for analysis of DNA and RNA sequence data | https://jgi.doe.gov/data-and-tools/bbtools/ | 61 |
| BCFtools | 1.9 | Utilities for variant calling and manipulating VCFs and BCFs | https://github.com/samtools/bcftools | 87 |
| Bedtools | 2.29.2 | A powerful tool set for genome arithmetic | https://github.com/arq5x/bedtools2 | 79 |
| BioPython | 1.76 | Tools for biological computation written in Python | https://github.com/biopython/biopython | 54 |
| BLAST+ | 2.9.0 | Basic local alignment search tool | https://blast.ncbi.nlm.nih.gov/Blast.cgi | 53 |
| Bowtie2 | 2.4.1 | A fast and sensitive gapped-read aligner | https://github.com/BenLangmead/bowtie2 | 88 |
| BWA | 0.7.17 | Burrows-Wheeler Aligner for short-read alignment | https://github.com/lh3/bwa/ | 77 |
| CD-HIT | 4.8.1 | Accelerated for clustering the next-generation sequencing data | https://github.com/weizhongli/cdhit | 55, 56 |
| CheckM | 1.1.2 | Assesses the quality of microbial genomes recovered from isolates, single cells, and metagenomes | https://github.com/Ecogenomics/CheckM | 72 |
| ClonalFrameML | 1.12 | Efficient inference of recombination in whole bacterial genomes | https://github.com/xavierdidelot/ClonalFrameML | 37 |
| DiagrammeR | 1.0.0 | Graph and network visualization using tabular data in R | https://github.com/rich-iannone/DiagrammeR | 89 |
| DIAMOND | 0.9.35 | Accelerated BLAST-compatible local sequence aligner | https://github.com/bbuchfink/diamond | 90 |
| eggNOG-Mapper | 2.0.1 | Fast genome-wide functional annotation through orthology assignment | https://github.com/eggnogdb/eggnog-mapper | 80, 81 |
| EMIRGE | 0.61.1 | Reconstructs full-length ribosomal genes from short-read sequencing data | https://github.com/csmiller/EMIRGE | 91 |
| FastANI | 1.3 | Fast whole-genome similarity (ANI) estimation | https://github.com/ParBLiSS/FastANI | 6 |
| FastTree 2 | 2.1.10 | Approximately-maximum-likelihood phylogenetic trees from alignments of nucleotide or protein sequences | http://www.microbesonline.org/fasttree | 92 |
| fastq-dl | 1.0.3 | Downloads FASTQ files from SRA or ENA repositories | https://github.com/rpetit3/fastq-dl | 58 |
| FastQC | 0.11.9 | A quality control analysis tool for high throughput sequencing data. | https://github.com/s-andrews/FastQC | 63 |
| fastq-scan | 0.4.3 | Outputs FASTQ summary statistics in JSON format | https://github.com/rpetit3/fastq-scan | 64 |
| FLASH | 1.2.11 | A fast and accurate tool to merge paired-end reads | https://ccb.jhu.edu/software/FLASH/ | 93 |
| freebayes | 1.3.2 | Bayesian haplotype-based genetic polymorphism discovery and genotyping | https://github.com/ekg/freebayes | 94 |
| GNU Parallel | 20200122 | A shell tool for executing jobs in parallel | https://www.gnu.org/software/parallel/ | 95 |
| GTDB-tk | 1.0.2 | A tool kit for assigning objective taxonomic classifications to bacterial and archaeal genomes | https://github.com/Ecogenomics/GTDBTk | 21 |
| HMMER | 3.3 | Biosequence analysis using profile hidden Markov models | http://hmmer.org/ | 23, 96, 97 |
| Infernal | 1.1.2 | Searches DNA sequence databases for RNA structure and sequence similarities | http://eddylab.org/infernal/ | 98 |
| IQ-TREE | 1.6.12 | Efficient phylogenomic software by maximum likelihood | https://github.com/Cibiv/IQ-TREE | 28 |
| ISMapper | 2.0 | Insertion sequence mapping software | https://github.com/jhawkey/IS_mapper | 82 |
| Lighter | 1.1.2 | Fast and memory-efficient sequencing error corrector | https://github.com/mourisl/Lighter | 62 |
| MAFFT | 7.455 | Multiple alignment program for amino acid or nucleotide sequences | https://mafft.cbrc.jp/alignment/software/ | 31 |
| Mash | 2.2.2 | Fast genome and metagenome distance estimation using MinHash | https://github.com/marbl/Mash | 17, 75 |
| Mashtree | 1.1.2 | Creates a tree using Mash distances | https://github.com/lskatz/mashtree | 83 |
| maskrc-svg | 0.5 | Masks recombination as detected by ClonalFrameML or Gubbins and draws an SVG | https://github.com/kwongj/maskrc-svg | 38 |
| McCortex | 1.0 | De novo genome assembly and multisample variant calling | https://github.com/mcveanlab/mccortex | 74 |
| MEGAHIT | 1.2.9 | Ultra-fast and memory-efficient (meta-)genome assembler | https://github.com/voutcn/megahit | 66 |
| MinCED | 0.4.2 | Mining CRISPRs in environmental data sets | https://github.com/ctSkennerton/minced | 99 |

mSystems®

**TABLE 1** (Continued)

| Name | Version | Description[a] | Link | Reference(s) |
|---|---|---|---|---|
| Minimap2 | 2.17 | A versatile pairwise aligner for genomic and spliced nucleotide sequences | https://github.com/lh3/minimap2 | 100 |
| ncbi-genome-download | 0.2.12 | Scripts to download genomes from the NCBI FTP servers | https://github.com/kblin/ncbi-genome-download | 35 |
| Nextflow | 19.10.0 | A DSL for data-driven computational pipelines | https://github.com/nextflow-io/nextflow | 3 |
| phyloFlash | 3.3b3 | Rapidly reconstruct the SSU rRNAs and explore phylogenetic composition of an Illumina (metagenomic data set) | https://github.com/HRGV/phyloFlash | 25 |
| Pigz | 2.3.4 | A parallel implementation of gzip for modern multiprocessor, multicore machines | https://zlib.net/pigz/ | 101 |
| Pilon | 1.23 | An automated genome assembly improvement and variant detection tool | https://github.com/broadinstitute/pilon/ | 69 |
| PIRATE | 1.0.3 | A toolbox for pan-genome analysis and threshold evaluation | https://github.com/SionBayliss/PIRATE | 84 |
| pplacer | 1.1.alpha19 | Phylogenetic placement and downstream analysis | https://github.com/matsen/pplacer | 24 |
| Prodigal | 2.6.3 | Fast, reliable protein-coding gene prediction for prokaryotic genomes | https://github.com/hyattpd/Prodigal | 22 |
| Prokka | 1.4.5 | Rapid prokaryotic genome annotation | https://github.com/tseemann/prokka | 36 |
| QUAST | 5.0.2 | Quality assessment tool for genome assemblies | http://quast.sourceforge.net/ | 71 |
| Racon | 1.4.13 | Ultrafast consensus module for raw de novo genome assembly of long uncorrected reads | https://github.com/lbcb-sci/racon | 102 |
| Roary | 3.13.0 | Rapid large-scale prokaryote pan genome analysis | https://github.com/sanger-pathogens/Roary | 7 |
| samclip | 0.2 | Filter SAM file for soft and hard clipped alignments | https://github.com/tseemann/samclip | 103 |
| SAMtools | 1.9 | Tools for manipulating next-generation sequencing data | https://github.com/samtools/samtools | 104 |
| Seqtk | 1.3 | A fast and lightweight tool for processing sequences in the FASTA or FASTQ format | https://github.com/lh3/seqtk | 105 |
| Shovill | 1.0.9se | Faster assembly of Illumina reads | https://github.com/tseemann/shovill | 65 |
| SKESA | 2.3.0 | Strategic k-mer extension for scrupulous assemblies | https://github.com/ncbi/SKESA | 67 |
| Snippy | 4.4.5 | Rapid haploid variant calling and core genome alignment | https://github.com/tseemann/snippy/ | 76 |
| SnpEff | 4.3.1 | Genomic variant annotations and functional effect prediction toolbox | http://snpeff.sourceforge.net/ | 106 |
| snp-dists | 0.6.3 | Pairwise SNP distance matrix from a FASTA sequence alignment | https://github.com/tseemann/snp-dists | 39 |
| SNP-sites | 2.5.1 | Rapidly extracts SNPs from a multi-FASTA alignment | https://github.com/sanger-pathogens/snp-sites | 107 |
| Sourmash | 3.2.0 | Compute and compare MinHash signatures for DNA data sets | https://github.com/dib-lab/sourmash | 19 |
| SPAdes | 3.13.0 | An assembly toolkit containing various assembly pipelines | https://github.com/ablab/spades | 26 |
| Trimmomatic | 0.39 | A flexible read trimming tool for Illumina NGS data | http://www.usadellab.org/cms/index.php?page=trimmomatic | 108 |
| Unicycler | 0.4.8 | Hybrid assembly pipeline for bacterial genomes | https://github.com/rrwick/Unicycler | 70 |
| vcf-annotator | 0.5 | Add biological annotations to variants in a VCF file | https://github.com/rpetit3/vcf-annotator | 109 |
| Vcflib | 1.0.0rc3 | A simple C++ library for parsing and manipulating VCF files | https://github.com/vcflib/vcflib | 110 |
| Velvet | 1.2.10 | Short read de novo assembler using de Bruijn graphs | https://github.com/dzerbino/velvet | 68 |
| VSEARCH | 2.14.1 | Versatile open-source tool for metagenomics | https://github.com/torognes/vsearch | 111 |
| vt | 2015.11.10 | A tool set for short-variant discovery in genetic sequence data | https://github.com/atks/vt | 112 |

[a]VCF, variant call format; BCF, binary variant call format; SVG, scalable vector graphics; JSON, JavaScript Object Notation; DSL, digital subscriber line; SSU, small subunit; NGS, next-generation sequencing.

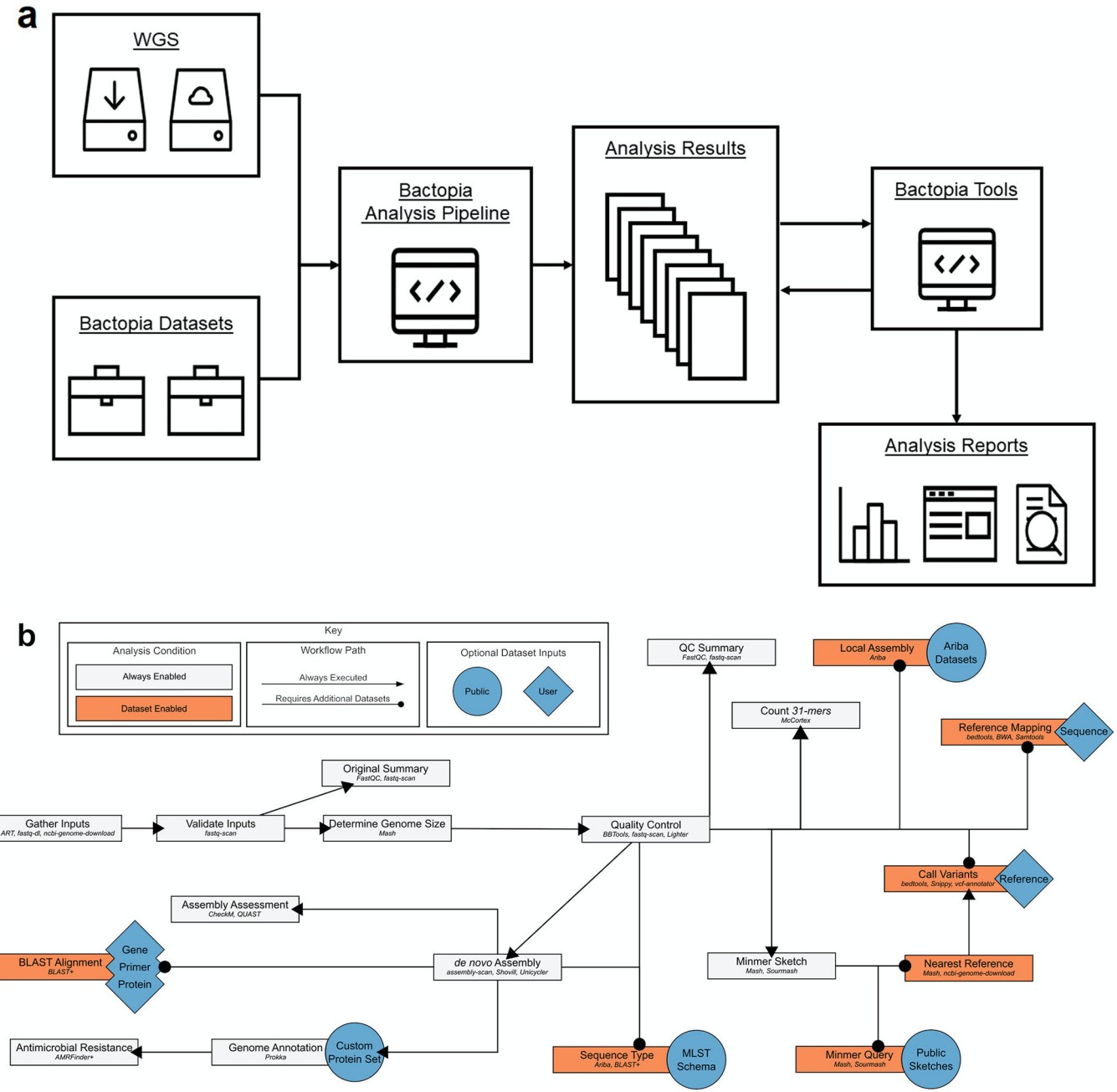

**FIG 1** Bactopia overview. (a) A general overview of the Bactopia workflow. (b) A detailed diagram of processing pathways within the Bactopia Analysis Pipeline showing optional data set inputs.

genomic workflow software programs that encompassed a similar range of functionality to Bactopia: ASA³P (8), TORMES (9), and the currently unpublished Nullarbor (10). The versions of these programs used many of the same component software programs (e.g., Prokka, SPAdes, BLAST+, and Roary) but differed in the philosophies underlying their design (Table 2). This made head-to-head runtime comparisons somewhat meaningless as each was aimed at a different analysis scenario and produced a different output. Bactopia was the most open-ended and flexible, allowing the user to customize input databases and providing a platform for downstream analysis by different BaTs rather than built-in pangenome and phylogeny creation. Bactopia also had some features not implemented in the other programs, such as SRA/ENA search and download and automated reference genome selection for identifying variants. Both Bactopia and ASA³P

**TABLE 2** A comparison of bacterial genome analysis workflows

| Feature | Bactopia | ASA³P | Nullarbor | TORMES |
|---|---|---|---|---|
| Version | 1.4.0 | 1.3.0 | 2.0.20191013 | 1.1 |
| Release date | 1 July 2020 | 2 May 2020 | 13 October 2019 | 14 April 2020 |
| Latest commit | 1 July 2020 | 26 June 2020 | 15 March 2020 | 28 May 2020 |
| Sequence technology | Illumina, Hybrid (Nanopore, Pacbio) | Illumina, Nanopore, PacBio | Illumina | Illumina |
| Single-end reads | Yes | Yes | No | No |
| Workflow | Nextflow | Groovy | Perl + Make | Bash |
| Resume if stopped | Yes | No | Yes | No |
| Reuse existing runs for expanded analysis | Yes | No | Yes | No |
| Built-in high-performance computing cluster and cloud capability | Yes | Yes | No | No |
| Individual program adjustable parameters | Yes | No | Yes | No |
| Batch processing from config file | Yes | Yes | Yes | Yes |
| Single sample processing from command line | Yes | No | Yes | No |
| Sequence depth downsample | Yes | No | Yes | No |
| Automatic reference selection for variant detection | Yes | No | No | No |
| Data download from SRA/ENA | Yes | No | No | No |
| Species identification | *k*-mers, 16S, ANI | *k*-mers, 16S, ANI | *k*-mers | *k*-mers |
| Comparative analysis | Separate process | Built-in process | Built-In Process | Built-in process |
| Summary | Text | HTML | HTML | R Markdown |
| Package manager | Bioconda | | Bioconda and Brew | Conda YAML |
| Container available | Yes | Yes | Yes | No |
| Documentation | Website | PDF manual | Readme | Readme |
| Github repository | https://github.com/bactopia/bactopia/ | https://github.com/oschwengers/asap | https://github.com/tseemann/nullarbor | https://github.com/nmquijada/tormes |

are highly scalable, and each can be seamlessly executed on local, cluster, and cloud environments with little effort required by the user. ASA³P was the only program to implement long-read assembly of multiple projects. TORMES was the only program to include a user-customizable RMarkdown for reporting and to have optional analyses specifically for *Escherichia* and *Salmonella*. Nullarbor was the only program to implement a prescreening method for filtering out potential biological outliers prior to full analysis.

**Use case: the *Lactobacillus* genus.** We performed a Bactopia analysis of publicly available raw Illumina data labeled as belonging to the *Lactobacillus* genus. *Lactobacillus* is an important component of the human microbiome, and cultured samples have been sequenced by several research groups over the past few years. *Lactobacillus crispatus* and other species are often the majority bacterial genus of the human vagina and are associated with low pH and reduction in pathogen burden (11). Samples of the genus are used in the food industry for fermentation in the production of yoghurt, kimchi, kombucha, and other common items. *Lactobacillus* is a common probiotic although recent genome-based transmission studies showed that bloodstream infections can follow after ingestion by immunocompromised patients (12).

In November 2019, we initiated Bactopia analysis using the following three commands:

```
# Build Lactobacillus dataset
bactopia datasets ~/bactopia-datasets --species 'Lactobacillus'
--cpus 10

# Query ENA for all Lactobacillus (tax id 1578) sequence
projects
bactopia search 1578 --prefix lactobacillus
# this creates a file called 'lactobacillus-accessions.txt'

# Process Lactobacillus samples
mkdir ~/bactopia
cd ~/bactopia
bactopia --accessions ~/lactobacillus-accessions.txt --datasets
```

 

**TABLE 3** Summary of *Lactobacillus* genome sequencing projects quality and coverage[a]

| Quality rank | No. of samples | Original coverage | Post-Bactopia coverage | Per-read quality score | Read length (bp) | Contig count | % of assembled genome size compared to estimated genome size |
|---|---|---|---|---|---|---|---|
| Gold | 967 | 213× | 100× | Q35 | 100 | 52 | 92 |
| Silver | 386 | 160× | 100× | Q35 | 100 | 110 | 93 |
| Bronze | 205 | 102× | 100× | Q34 | 100 | 90 | 93 |
| Exclude | 48 | 26× | 22× | Q34 | 100 | 706 | 93 |
| Unprocessed | 58 | | | | | | |

[a]All values except number of samples are medians.

```
~/bactopia-datasets --species lactobacillus --coverage 100
--cpus 4 --min_genome_size 1000000 --max_genome_size 4200000
```

The "bactopia datasets" subcommand automated downloading of BaDs. With these parameters, we downloaded and formatted the following data sets: Ariba (13) reference databases for the Comprehensive Antibiotic Resistance Database (CARD) and the core Virulence Factor Database (VFDB) (14, 15), RefSeq Mash sketch (16, 17), GenBank Sourmash signatures (18, 19), PLSDB BLAST database and Mash sketch (20), and a clustered protein set and Mash sketch from completed *Lactobacillus* genomes (`--species 'lactobacillus'`) available from NCBI Assembly (RefSeq). This took 25 min to complete.

The "bactopia search" subcommand produced a list of accession numbers for 2,030 experiments that had been labeled as "*Lactobacillus*" (taxonomy identifier [taxon ID]: 1578) (Data Set S1). After filtering for only Illumina sequencing, 1,664 accession numbers for experiments remained (Data Set S2).

The main "bactopia" command automated BaAP processing of the list of accessions (`--accessions ~/lactobacillus-accessions.txt`) using the downloaded BaTs (`--datasets ~/bactopia-datasets --species lactobacillus`). Here, we chose a standard maximum coverage per genome of 100× (`--coverage 100`), based on the estimated genome size. We used the range of genome sizes (1.2 Mb to 3.7 Mb) for the completed *Lactobacillus* genomes to require that the estimated genome size for each sample be between 1 Mbp (`--min_genome_size 1000000`) and 4.2 Mbp (`--max_genome_size 4200000`).

Samples were processed on a 96-core SLURM cluster with 512 GB of available RAM. Analysis took approximately 2.5 days to complete, with an estimated runtime of 30 min per sample (determined by adding up the median process runtime, for 17 different processes in total, in BaAP). No individual process used more than 8 GB of memory, with all but five using less than 1 GB. Nextflow (3) recorded detailed statistics on resource usage, including CPU, memory, job duration, and input-output (I/O). (Data Set S3).

**Analysis of *Lactobacillus* genomes using BaTs.** The BaAP outputted a directory of directories named after the unique experiment accession number for each sample. Within each sample directory were subdirectories for the output of each analysis run. These data structures were recognized by BaTs for subsequent analysis.

We used BaT "summary" to generate a summary report of our analysis. The report includes an overview of sequence quality, assembly statistics, and predicted antimicrobial resistances and virulence factors. It also outputs a list of samples that fail to meet minimum sequencing depth and/or quality thresholds.

```
bactopia  tools  summary  --bactopia  ~/bactopia  --prefix
lactobacillus
# this creates a file called 'lactobacillus-exclude.txt'
```

BaT "summary" grouped samples as gold, silver, bronze, exclude, or unprocessed, based on BaAP completion, minimum sequencing coverage, per-read sequencing mean quality, minimum mean read length, and assembly quality (Table 3; Fig. S2). To be placed in a group, a sample had to meet each cutoff. Cutoffs were based on those used by the Staphopia Analysis Pipeline (StAP) (4) with the addition of a contig count

cutoff. For this analysis we used the default values for these cutoffs to group our samples. Gold samples were defined as those having greater than 100× coverage, per-read mean quality greater than Q30, mean read length greater than 95 bp, and an assembly with fewer than 100 contigs. Silver samples were defined as those having greater than 50× coverage, per-read mean quality greater than Q20, mean read length greater than 75 bp, and an assembly with less than 200 contigs. Bronze samples were defined as those having greater than 20× coverage, per-read mean quality greater than Q12, mean read length greater than 49 bp, and an assembly with fewer than 500 contigs. A total of 106 samples (the exclude and unprocessed groups) were excluded from further analysis (Table S1). Forty-eight samples that failed to meet the minimum thresholds for bronze quality were assigned to the exclude group. Fifty-eight samples that were not processed by BaAP due to sequencing-related errors or because of the estimated genome sizes were grouped as unprocessed. Of these, one (SRA accession no. SRX4526092) was labeled as paired end but did not have both sets of reads, one (SRA accession no. SRX1490246) was identified to be an assembly converted to FASTQ format, and 14 had insufficient sequencing depth. The remaining 42 samples, unprocessed by BaAP, had an estimated genome size which exceeded 4.2 Mbp (set at runtime). We queried these samples against available GenBank and RefSeq sketches using Mash screen and Sourmash lca gather. There were 36 samples that contained evidence for *Lactobacillus* but also sequences for other bacterial species, phage, virus, and plant genomes. There were six samples that contained no evidence for *Lactobacillus*, four of which had matches to multiple bacterial species, and two of which had matches only to *Saccharomyces cerevisiae*.

There were 1,558 samples with gold, silver, or bronze quality (Table 3) that were used for further analysis. For these we found that, on average, the assembled genome size was about 12% smaller than the estimated genome size (Table 3; Fig. S3). If we assume that the assembled genome size is a better indicator of a sample's genome size, the average coverage before quality control (QC) increased from 220× to 268×. In this use case, the *Lactobacillus* genus, it was necessary to estimate genome sizes, but in dealing with samples from a single species, it may be better to provide a known genome size.

For visualization of the phylogenetic relationships of the samples, we used the "phyloflash" and "gtdb" BaTs.

```
bactopia tools phyloflash --phyloflash ~/bactopia-datasets/
16s/138 --bactopia ~/bactopia --cpus 16 --exclude ~/bactopia-
tool/summary/lactobacillus-exclude.txt
```

```
bactopia tools gtdb --gtdb ~/bactopia-datasets/gtdb/db
--bactopia ~/bactopia --cpus 48 --exclude ~/bactopia-tool/
summary/lactobacillus-exclude.txt
```

The "gtdb" BaT used GTDB-Tk (21) to assign a taxonomic classification to each sample. GTDB-Tk used the assembly to predict genes with Prodigal (22), identify GTDB marker genes (5) (`--gtdb ~/bactopia-datasets/gtdb/db`) for phylogenetic inference with HMMER3 (23), and find the maximum-likelihood placement of each sample on the GTDB-Tk reference tree with pplacer (24). A taxonomic classification was assigned to 1,554 samples, and 4 samples failed classification due to insufficient marker gene coverage or marker genes with multiple hits.

The "phyloflash" BaT used the phyloFlash tool (25) to reconstruct a 16S rRNA gene from each sample that was used for phylogenetic reconstruction (Fig. 2). Samples that failed to meet quality cutoffs were excluded from this analysis (`--exclude ~/bactopia-tool/summary/lactobacillus-exclude.txt`). The 16S rRNA was reconstructed from a SPAdes (26) assembly and annotated against the SILVA (27) rRNA database (`v138, --phyloflash ~/bactopia-datasets/16s/138`) for 1,470 samples. There were 88 samples that were excluded from the phylogeny: 12 samples that did not meet the requirement of a mean read length of

mSystems®

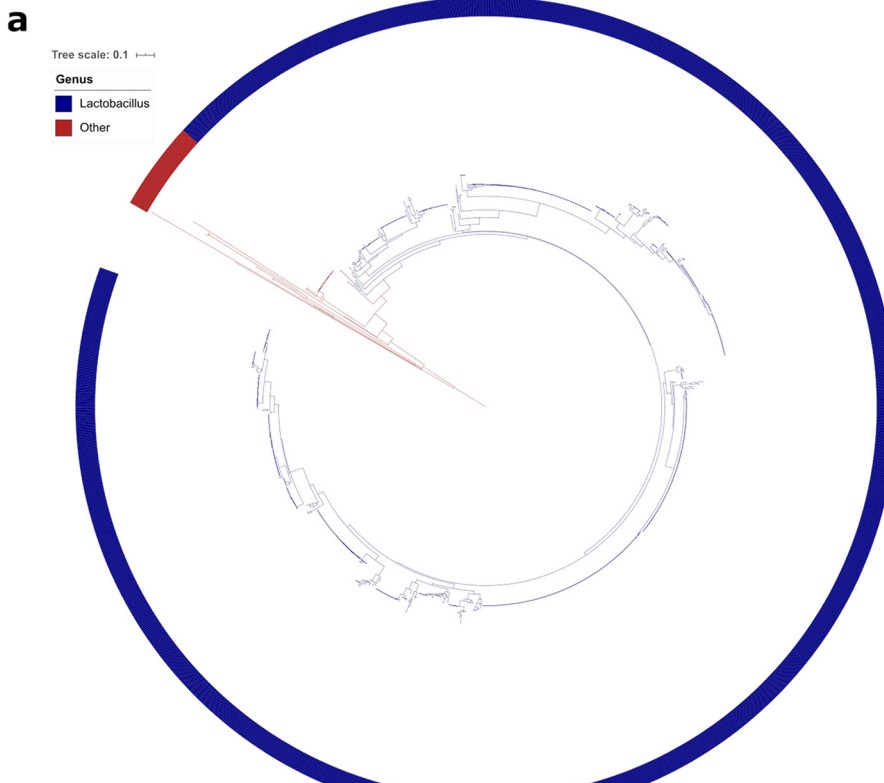

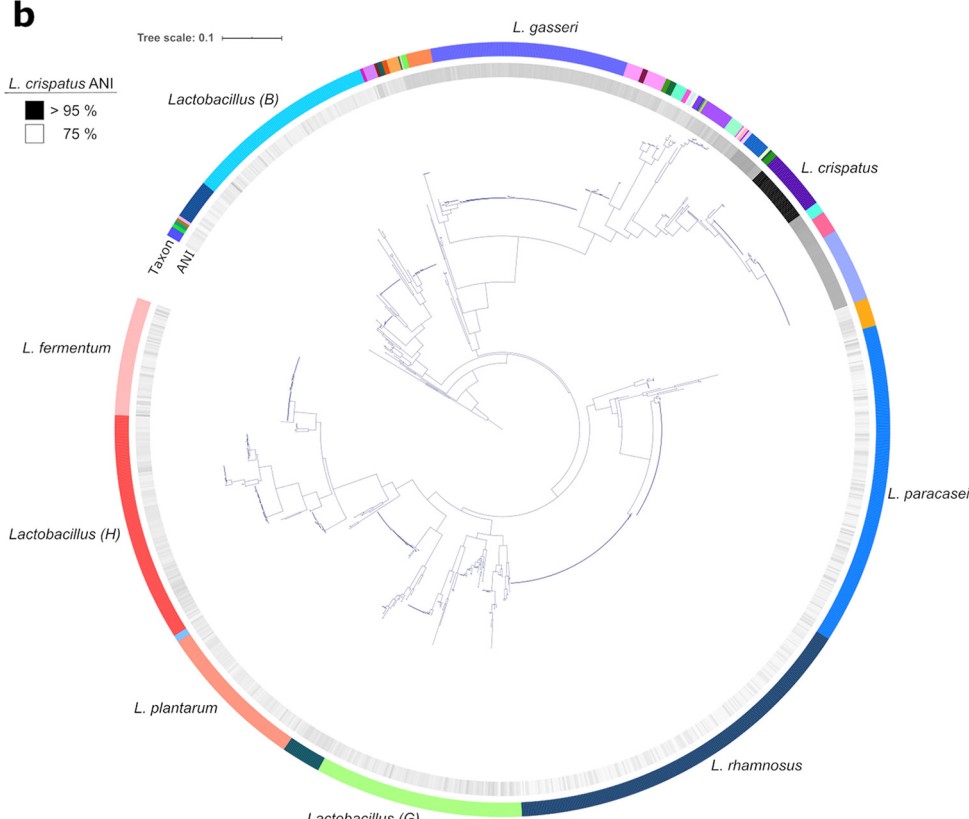

**FIG 2** Maximum-likelihood phylogeny from reconstructed 16S rRNA genes. A phylogenetic representation of 1,470 samples using IQ-Tree (28–30). (a) A tree of the full set of samples. The outer ring represents the genus assigned

50 bp, 17 samples in which a 16S gene could not be reconstructed, 19 samples that had a mismatch in assembly and mapped-read taxon designations, and 40 samples that had 16S genes reconstructed for multiple species. A phylogenetic tree was created with IQ-TREE (28–30) based on a multiple-sequence alignment of the reconstructed 16S genes with MAFFT (31). Taxonomic classifications from GTDB-Tk were used to annotate the 16S genes with iTOL (32).

A recent analysis of completed genomes in the NCBI found 239 discontinuous *de novo Lactobacillus* species using a 94% ANI cutoff (33). Based on GTDB taxonomic classification, which applies a 95% ANI cutoff, we identified 161 distinct *Lactobacillus* species in 1,554 samples. The five most sequenced *Lactobacillus* species, accounting for 45% of the total, were *L. rhamnosus* ($n = 225$), *L. paracasei* ($n = 180$), *L. gasseri* ($n = 132$), *L. plantarum* ($n = 86$), and *L. fermentum* ($n = 80$). Within these five species the assembled genomes sizes were remarkably consistent (Fig. S4). There were 58 samples that were not classified as *Lactobacillus*, of which 34 were classified as *Streptococcus pneumoniae* by both 16S gene sequencing and GTDB (Table S2).

We found that 505 (~33%) of 1,554 taxonomic classifications by 16S gene and GTDB were in conflict with the taxonomy according to the NCBI SRA, illustrating the importance of an unbiased approach to understanding sample context. In samples that had both a 16S and GTDB taxonomic classification, there was disagreement in 154 out of 1,467 samples. Of these, 47% were accounted for by the recently described *L. paragasseri* (34) ($n = 72$). This possibly highlights a lag in the reclassification of assemblies in the NCBI Assembly database.

Analysis of the pangenome of the entire genus using a tool such as Roary (7) would return only a few core genes, owing to sequence divergence of evolutionarily distant species. However, because the "roary" BaT can be supplied with a list of individual samples, it is possible to isolate the analysis to the species level. As an example of using BaTs to focus on a particular group within the larger set of results, we chose *L. crispatus*, a species commonly isolated from the human vagina and also found in the guts/feces of poultry.

```
bactopia  tools  fastani  --bactopia  ~/bactopia  --exclude
~/bactopia-tool/summary/lactobacillus-exclude.txt    --accession
GCF_003795065.1 --refseq_only --minFraction 0.0

# Identify samples with > 95% ANI to L. crispatus

awk '{if ($3 > 95){print $0}}' ~/bactopia-tool/fastani/fastani
.tsv | grep "RX" > ~/crispatus-include.txt

bactopia tools roary --bactopia ~/bactopia --cpus 20 --include
~/crispatus-include.txt --species "lactobacillus crispatus" --n
```

We used the "fastani" BaT to estimate the ANI of all samples against a single (`--refseq_only`) randomly selected *L. crispatus* completed genome (NCBI Assembly accession no. GCF_003795065; `--accession`) with FastANI (6). A cutoff of greater than 95% ANI was used to categorize a sample as *L. crispatus*. A pan-genome analysis was conducted on only the samples categorized as *L. crispatus* (`--include ~/crispatus-include.txt`) using the "roary" BaT. The "roary" BaT downloaded all available completed *L. crispatus* genomes with ncbi-genome-download (35), formatted the completed genomes with Prokka (36), created a pan-genome and core-genome alignment (`--n`) with Roary (7), identified and masked recombination with Clonal-

**FIG 2** Legend (Continued)
by GTDB-Tk, as indicated. (b) The same tree as shown in panel a, but with the non-*Lactobacillus* clade collapsed. Major groups of *Lactobacillus* species (indicated with a letter) and the most sequenced *Lactobacillus* species have been labeled. The inner ring represents the average nucleotide identity (ANI), determined by FastANI (6), of samples to *L. crispatus*. The tree was built from a multiple-sequence alignment (31) of 16S genes reconstructed by phyloFlash (25) with 1,281 parsimony-informative sites. The likelihood score for the consensus tree constructed from 1,000 bootstrap trees was −54,698. Taxonomic classifications were assigned by GTDB-Tk (21).

**TABLE 4** *Lactobacillus crispatus* genomes used in pan-genome analysis[a]

| Accession no.[b] | | | | | |
| --- | --- | --- | --- | --- | --- |
| BioProject | BioSample | Experiment[b] | Host[c] | Source[c] | Reference |
| PRJEB8104 | SAMEA3319334 | ERX1126086 | Human* | Urine* | |
| | SAMEA3319350 | ERX1126089 | Human* | Urine* | |
| | SAMEA3319265 | ERX1126106 | Human* | Urine* | |
| | SAMEA3319366 | ERX1126138 | Human* | Urine* | |
| | SAMEA3319373 | ERX1126140 | Human* | Urine* | |
| | SAMEA3319383 | ERX1126143 | Human* | Urine* | |
| | SAMEA3319392 | ERX1126150 | Human* | Urine* | |
| PRJEB22112 | SAMEA104208649 | ERX2150228 | Human* | Urine* | |
| | SAMEA104208650 | ERX2150229 | Human* | Urine* | |
| PRJEB3060 | SAMEA1920319 | ERX271950 | Human* | Unknown | |
| | SAMEA1920326 | ERX271958 | Human* | Unknown | |
| | SAMEA1920319 | ERX450852 | Human* | Unknown | |
| | SAMEA1920326 | ERX450860 | Human* | Unknown | |
| PRJNA50051 | SAMN00109860 | SRX026143 | Human* | Vaginal* | |
| PRJNA272101 | SAMN03854351 | SRX1090887 | Human | Urine | 113 |
| PRJNA50053 | SAMN00829399 | SRX130900 | Human* | Vaginal* | |
| PRJNA50057 | SAMN00829123 | SRX130912 | Human* | Vaginal* | |
| PRJNA50067 | SAMN00829125 | SRX130914 | Human* | Vaginal* | |
| PRJNA52107 | SAMN01057066 | SRX155504 | Human* | Vaginal* | |
| PRJNA52105 | SAMN01057067 | SRX155505 | Human* | Vaginal* | |
| PRJNA52107 | SAMN01057066 | SRX155863 | Human* | Vaginal* | |
| PRJNA52105 | SAMN01057067 | SRX155875 | Human* | Vaginal* | |
| PRJNA379934 | SAMN06624125 | SRX2660270 | Human | Eye | |
| PRJNA222257 | SAMN02369387 | SRX456245 | Human | Eye | |
| PRJNA231221 | SAMN11056458 | SRX5949263 | Human | Vaginal | 114 |
| PRJNA547620 | SAMN11973370 | SRX5986001 | Human | Vaginal | 115 |
| | SAMN11973369 | SRX5986002 | Human | Vaginal | 115 |
| | SAMN11973371 | SRX5986003 | Human | Vaginal | 115 |
| PRJNA557339 | SAMN12395213 | SRX6613945 | Human | Vaginal | 116 |
| PRJNA563077 | SAMN12667791 | SRX6959881 | Human | Gut | 40 |
| | SAMN12667801 | SRX6959883 | Chicken | Gut | 40 |
| | SAMN12667803 | SRX6959885 | Human | Gut | 40 |
| | SAMN12667804 | SRX6959886 | Turkey | Gut | 40 |
| | SAMN12667805 | SRX6959887 | Human | Eye | 40 |
| | SAMN12667793 | SRX6959888 | Chicken | Gut | 40 |
| | SAMN12667794 | SRX6959889 | Chicken | Gut | 40 |
| | SAMN12667795 | SRX6959890 | Chicken | Gut | 40 |
| | SAMN12667796 | SRX6959891 | Chicken | Gut | 40 |
| | SAMN12667797 | SRX6959892 | Chicken | Gut | 40 |
| | SAMN12667798 | SRX6959893 | Chicken | Gut | 40 |
| | SAMN12667799 | SRX6959894 | Chicken | Gut | 40 |
| | SAMN12667800 | SRX6959895 | Chicken | Gut | 40 |
| PRJNA531669 | SAMN11372136 | GCF_009769205 | Chicken | Gut | 117 |
| PRJNA231221 | SAMN11056458 | GCF_009730275 | Human | Vaginal | 114 |
| PRJNA431864 | SAMN08409124 | GCF_003971565 | Human | Vaginal | 118 |
| PRJNA499123 | SAMN10343598 | GCF_003795065 | Human | Vaginal | 119 |

[a]*Lactobacillus crispatus* samples ($n = 42$) were used in the pan-genome analysis.
[b]NCBI Assembly (beginning with GCF) or SRA experiment accession number.
[c]The host and source were collected from metadata associated with the BioSample or available publications. In cases when a host and/or source was not explicitly stated, it was inferred from available metadata (denoted by an asterisk).

FrameML (37) and maskrc-svg (38), and created a phylogenetic tree with IQ-TREE (28–30) and a pairwise SNP distance matrix with snp-dists (39).

ANI analysis revealed 38 samples as having >96.1% ANI to *L. crispatus*, with no other sample greater than 83.1%. Four completed *L. crispatus* genomes were also included in the analysis (Table 4), for a total of 42 genomes. The pan-genome of *L. crispatus* was revealed to have 7,037 gene families and 972 core genes (Fig. 3). Similar to a recent analysis by Pan et al. (40), *L. crispatus* was separated into two main phylogenetic groups, one associated with human vaginal isolates and the other having more mixed provenance and including chicken, turkey, and human gut isolates.

Last, we looked at patterns of antibiotic resistance across the genus using a table, generated by the "summary" BaT, of resistance genes and loci called by AMRFinder+ (41).

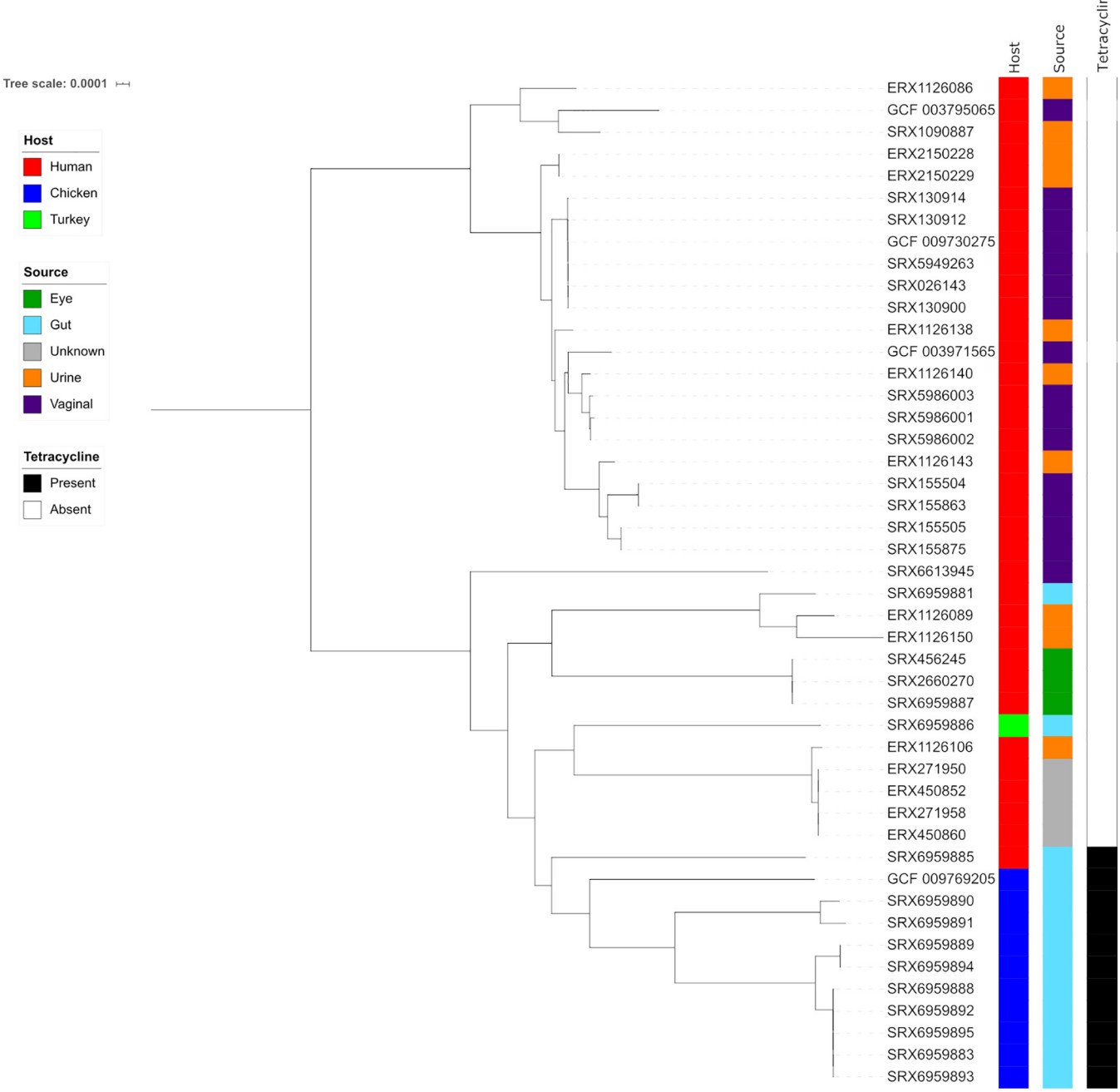

**FIG 3** Core-genome maximum-likelihood phylogeny of *Lactobacillus crispatus*. A core-genome phylogenetic representation using IQ-Tree (28–30) of 42 *L. crispatus* samples. The putatively recombinant positions predicted using ClonalFrameML (37) were removed from the alignment with maskrc-svg (38). The tree was built from 972 core genes identified by Roary with 9,209 parsimony-informative sites. The log-likelihood score for the consensus tree constructed from 1,000 bootstrap trees was −1,418,106.

Only 79 out of 1,496 *Lactobacillus* samples defined by GTDB-Tk (21) were found to have predicted resistance using AMRFinder+. The most common resistance categories were tetracyclines (67 samples), followed by macrolides, lincosamides, and aminoglycosides (16, 15, and 11 samples, respectively). Species with the highest proportion of resistance included *L. amylovorus* (12/14 tetracycline resistant) and *L. crispatus* (10/42 tetracycline resistant). Only three genomes of *L. amylophilus* were included in the study, but each contained matches to genes for macrolide, lincosamide, and tetracycline resistance. The linking thread between these species is that they are each commonly isolated from

agricultural animals. The high proportion of *L. crispatus* samples isolated from chickens that were tetracycline resistant has been previously observed (42, 43) (Fig. 3).

A recent analysis of 184 *Lactobacillus* type strain genomes by Campedelli et al. (44) found a higher percentage of type strains with aminoglycoside (20/184), tetracycline (18/184), erythromycin (6/184), and clindamycin (60/184) resistance. Forty-two of the type strains had chloramphenicol resistance genes whereas, here, AMRFinder+ returned only 1/1,467 genes. These differences probably reflect a combination of the different sampling biases of the studies and the strategy of Campedelli et al. to use a relaxed threshold for hits to maximize sensitivity (blastp matches against the CARD database with acid sequence identity of 30% and query coverage of 70% [44]). Resistance is probably undercalled by both methods because of a lack of well-characterized resistance loci from the *Lactobacillus* genus to use for comparison.

## DISCUSSION

Bactopia is a flexible workflow for bacterial genomics. It can be run on a laptop for a single bacterial sample, but, critically, the underlying Nextflow framework allows it to make efficient use of large clusters and cloud-computing environments to process the many thousands of genomes that are currently being generated. For users that are not familiar with bacterial genomic tools and/or who require a standardized pipeline, Bactopia is a one-stop shop that can be easily deployed using conda, Docker, and Singularity containers. For researchers with particular interest in individual species or genera, BaDs can be highly customized with taxon-specific databases.

The current version of Bactopia has only minimal support for long-read data, but this is an area that we plan to expand in the future. We also plan to implement more comparative analyses in the form of additional BaTs. With a framework set in place for developing BaTs, it should be possible to make a toolbox of workflows that not only can be used for all bacteria but are also customized for annotating genes and loci specific for particular species.

## MATERIALS AND METHODS

**Bactopia Data Sets.** The Bactopia pipeline can be run without downloading and formatting Bactopia Data Sets (BaDs). However, providing them enriches the downstream analysis. Bactopia can import specific existing public data sets, as well as accessible user-provided data sets in the appropriate format. A subcommand ("bactopia datasets") was created to automate downloading, building, and (or) configuring these data sets for Bactopia.

BaDs can be grouped into those that are general and those that are user supplied. General data sets include a Mash (17) sketch of the NCBI RefSeq (16) and PLSDB (20) databases and a Sourmash (19) signature of microbial genomes (including viral and fungal) from the NCBI GenBank (18) database. Ariba (13), a software program for detecting genes in raw read (FASTQ) files, uses a number of default reference databases for virulence and antibiotic resistance. The available Ariba data sets include ARG-ANNOT (45), CARD (15), MEGARes (46), NCBI Reference Gene Catalog (47), plasmidfinder (48), resfinder (49), SRST2 (50), VFDB (14), and VirulenceFinder (51).

When an organism name is provided, additional data sets are set up. If a multilocus sequence typing (MLST) schema is available for the species, it is downloaded from PubMLST.org (52) and set up for BLAST+ (53) and Ariba. Each RefSeq completed genome for the species is downloaded using ncbi-genome-download (35). A Mash sketch is created from the set of downloaded completed genomes to be used for automatic reference selection for variant calling. Protein sequences are extracted from each genome with BioPython (54), clustered using CD-HIT (55, 56), and formatted to be used by Prokka (36) for annotation. Users may also provide their own organism-specific reference data sets to be used for BLAST+ alignment, short-read alignment, or variant calling.

**Bactopia Analysis Pipeline.** The Bactopia Analysis Pipeline (BaAP) takes input FASTQ or preassembled genomes as FASTA files and optional user-specified BaDs and performs a number of workflows that are based on either *de novo* whole-genome assembly, reference mapping, or sequence decomposition (i.e., *k*-mer-based approaches) (Fig. 1b). BaAP has incorporated numerous existing bioinformatic tools (Table 1) into its workflow (Fig. 1b; see also Fig. S1 in the supplemental material). For each tool, many of the input parameters are exposed to the user, allowing for fine-tuning analysis.

**BaAP: acquiring FASTQs.** Bactopia provides multiple ways for users to provide their FASTQ-formatted sequences. Input FASTQs can be local or downloaded from public repositories or preassembled genomes as FASTA files. There is also an option for hybrid assembly of Illumina and long-read data.

Local sequences can be processed one at a time or in batches. To process a single sample, the user provides the path to the FASTQ(s) and a sample name. For multiple samples, this method does not make efficient use of Nextflow's queue system. Alternatively, users can provide a "file of filenames" (FOFN),

which is a tab-delimited file with information about samples and paths to the corresponding FASTQ(s). By using the FOFN method, Nextflow queues each sample and makes efficient use of available resources. A subcommand ("bactopia prepare") was created to automate the creation of an FOFN.

Raw sequences available from public repositories (e.g., European Nucleotide Archive [ENA], Sequence Read Archive [SRA], DNA Data Bank of Japan [DDBJ], or NCBI Assembly) can also be processed by Bactopia. Sequences associated with a provided experiment accession number (e.g., DRX, ERX, or SRX prefix) or NCBI Assembly accession number (e.g., GCF or GCA prefix) are downloaded and processed exactly as local sequences would be. A subcommand ("bactopia search") was created which allows users to query ENA to create a list of experiment accession numbers from the ENA Data Warehouse API (57) associated with a BioProject accession number, taxon ID, or organism name.

**BaAP: validating FASTQs.** The path for input FASTQ(s) is validated, and, if necessary, sequences from public repositories are downloaded using fastq-dl (58). If a preassembled genome is provided as an input, 2- by 250-bp paired-end reads are simulated using ART (59). Once validated, the FASTQ input(s) is tested to determine if it meets a minimum threshold for continued processing. All BaAP steps expect to use Illumina sequence data, which represent the great majority of genome projects currently generated. FASTQ files that are explicitly marked as non-Illumina or have properties that suggest that they are non-Illumina (e.g., read length or error profile) are excluded. By default, input FASTQs must exceed 2,241,820 bases (20× coverage of the smallest bacterial genome, *Nasuia deltocephalinicola* [60]) and 7,472 reads (minimum required base pairs/300 bp, the longest available reads from Illumina). If estimated, the genome size must be between 100,000 bp and 18,040,666 bp, which is based on the range of known bacterial genome sizes (*N. deltocephalinicola*, NCBI accession no. GCF_000442605, 112,091 bp; *Minicystis rosea*, NCBI accession no. GCF_001931535, 16,040,666 bp). Failure to pass these requirements excludes the samples from further subsequent analysis. The threshold values can be adjusted by the user at runtime.

**BaAP: FastQ quality control and generation of pFASTQ.** Input FASTQs that pass the validation steps undergo quality control steps to remove poor-quality reads. BBDuk, a component of BBTools (61), removes Illumina adapters and phiX contaminants and filters reads based on length and quality. Base calls are corrected using Lighter (62). At this stage, the default procedure is to downsample the FASTQ file to an average 100× genome coverage (if over 100×) with Reformat (from BBTools). This step, which was used in StAP (4), significantly saves computing time at little final cost to assembly or SNP calling accuracy. The genome size for coverage calculation is either provided by the user or estimated based on the FASTQ data by Mash (17). The user can provide their own value for downsampling FASTQs or disable it completely. Summary statistics before and after QC are created using FastQC (63) and fastq-scan (64). After QC, the original FASTQs are no longer used, and only the processed FASTQs (pFASTQ) are used in subsequent analysis.

**BaAP: assembly, reference mapping, and decomposition.** BaAP uses Shovill (65) to create a draft *de novo* assembly with MEGAHIT (66), SKESA (67) (default), SPAdes (26), or Velvet (68) and makes corrections using Pilon (69) from the pFASTQ. Alternatively, if long reads were provided with paired-end pFASTQ, a hybrid assembly is created with Unicycler (70). The quality of the draft assembly is assessed by QUAST (71) and CheckM (72). Summary statistics for the draft assembly are created using assembly scan (73). If the total size of the draft assembly fails to meet a user-specified minimum size, further assembly-based analyses are discontinued. Otherwise, a BLAST+ (53) nucleotide database is created from the contigs. The draft assembly is also annotated using Prokka (36). If available at runtime, Prokka will first annotate with a clustered RefSeq protein set, followed by its default databases. The annotated genes and proteins are then subjected to antimicrobial resistance prediction with AMRFinder+ (47).

For each pFASTQ, sketches are created using Mash ($k = 21,31$) and Sourmash (19) ($k = 21,31,51$). McCortex (74) is used to count 31-mers in the pFASTQ.

**BaAP: optional steps.** At runtime, Bactopia checks for BaDs specified by the command line (if any) and adjusts the settings of the pipeline accordingly. Examples of processes executed only if a BaDs is specified include Ariba (13) analysis for each available reference data set, sequence containment estimation against RefSeq (16) with mash screen (75) and against GenBank (18) with sourmash lca gather (19), and PLSDB (20), with mash screen and BLAST+. The sequence type (ST) of the sample is determined with BLAST+ and Ariba. The nearest reference RefSeq genome, based on mash (17) distance, is downloaded with ncbi-genome-download (35), and variants are called with Snippy (76). Alternatively, one or more reference genomes can be provided by the user. Users can also provide sequences for sequence alignment with BLAST+ and per-base coverage with BWA (77, 78) and Bedtools (79).

**Bactopia tools.** After BaAP has successfully finished, it will create a directory for each strain with subdirectories for each analysis result. The directory structure is independent of the project or options chosen. Bactopia Tools (BaTs) are a set of comparative-analysis workflows written using Nextflow that take advantage of the predictable output structure from BaAP. Each BaT is created from the same framework and a subcommand ("bactopia tools create") is available to simplify the creation of future BaTs.

Five BaTs were used for analyses in this article. The "summary" BaT outputs a summary report of the set of samples and a list of samples that failed to meet thresholds set by the user. This summary includes basic sequence and assembly stats as well as technical (pass/fail) information. The "roary" BaT creates a pan-genome of the set of samples with Roary (7), with the option to include RefSeq (16) completed genomes. The "fastani" BaT determines the pairwise average nucleotide identity (ANI) for each sample with FastANI (6). The "phyloflash" BaT reconstructs 16S rRNA gene sequences with phyloFlash (25). The "gtdb" BaT assigns taxonomic classifications from the Genome Taxonomy Database (GTDB) (5) with GTDB-tk (21). Each Bactopia tool has a separate Nextflow workflow with its own conda environment,

Docker image, and Singularity image. Additional BaTs are currently available for eggNOG-mapper (80, 81), ISMapper (82), Mashtree (83), and PIRATE (84).

**Data availability.** Raw Illumina sequences of *Lactobacillus* samples used in this study were acquired from experiments submitted under BioProject accession numbers PRJDB1101, PRJDB1726, PRJDB4156, PRJDB4955, PRJDB5065, PRJDB5206, PRJDB6480, PRJDB6495, PRJEB10572, PRJEB11980, PRJEB14693, PRJEB18589, PRJEB19875, PRJEB21025, PRJEB21680, PRJEB22112, PRJEB22252, PRJEB23845, PRJEB24689, PRJEB24698, PRJEB24699, PRJEB24700, PRJEB24701, PRJEB24713, PRJEB24715, PRJEB25194, PRJEB2631, PRJEB26638, PRJEB2824, PRJEB29398, PRJEB29504, PRJEB2977, PRJEB3012, PRJEB3060, PRJEB31213, PRJEB31289, PRJEB31301, PRJEB31307, PRJEB5094, PRJEB8104, PRJEB8721, PRJEB9718, PRJNA165565, PRJNA176000, PRJNA176001, PRJNA183044, PRJNA184888, PRJNA185359, PRJNA185406, PRJNA185584, PRJNA185632, PRJNA185633, PRJNA188920, PRJNA188921, PRJNA212644, PRJNA217366, PRJNA218804, PRJNA219157, PRJNA222257, PRJNA224116, PRJNA227106, PRJNA227335, PRJNA231221, PRJNA234998, PRJNA235015, PRJNA235017, PRJNA247439, PRJNA247440, PRJNA247441, PRJNA247442, PRJNA247443, PRJNA247444, PRJNA247445, PRJNA247446, PRJNA247452, PRJNA254854, PRJNA255080, PRJNA257137, PRJNA257138, PRJNA257139, PRJNA257141, PRJNA257142, PRJNA257182, PRJNA257185, PRJNA257853, PRJNA257876, PRJNA258355, PRJNA258500, PRJNA267549, PRJNA269805, PRJNA269831, PRJNA269832, PRJNA269860, PRJNA269905, PRJNA270961, PRJNA270962, PRJNA270963, PRJNA270964, PRJNA270965, PRJNA270966, PRJNA270967, PRJNA270968, PRJNA270969, PRJNA270970, PRJNA270972, PRJNA270973, PRJNA270974, PRJNA272101, PRJNA272102, PRJNA283920, PRJNA289613, PRJNA29003, PRJNA291681, PRJNA296228, PRJNA296248, PRJNA296274, PRJNA296298, PRJNA296309, PRJNA296751, PRJNA296754, PRJNA298448, PRJNA299992, PRJNA300015, PRJNA300023, PRJNA300088, PRJNA300119, PRJNA300123, PRJNA300179, PRJNA302242, PRJNA303235, PRJNA303236, PRJNA305242, PRJNA306257, PRJNA309616, PRJNA312743, PRJNA315676, PRJNA316969, PRJNA322958, PRJNA322959, PRJNA322960, PRJNA322961, PRJNA336518, PRJNA342061, PRJNA342757, PRJNA347617, PRJNA348789, PRJNA376205, PRJNA377666, PRJNA379934, PRJNA381357, PRJNA382771, PRJNA388578, PRJNA392822, PRJNA397632, PRJNA400793, PRJNA434600, PRJNA436228, PRJNA474823, PRJNA474907, PRJNA476494, PRJNA477598, PRJNA481120, PRJNA484967, PRJNA492883, PRJNA493554, PRJNA496358, PRJNA50051, PRJNA50053, PRJNA50055, PRJNA50057, PRJNA50059, PRJNA50061, PRJNA50063, PRJNA50067, PRJNA50115, PRJNA50117, PRJNA50125, PRJNA50133, PRJNA50135, PRJNA50137, PRJNA50139, PRJNA50141, PRJNA50159, PRJNA50161, PRJNA50163, PRJNA50165, PRJNA50167, PRJNA50169, PRJNA50173, PRJNA504605, PRJNA504734, PRJNA505088, PRJNA52105, PRJNA52107, PRJNA52121, PRJNA525939, PRJNA530250, PRJNA533291, PRJNA533837, PRJNA542049, PRJNA542050, PRJNA542054, PRJNA543187, PRJNA544527, PRJNA547620, PRJNA552757, PRJNA554696, PRJNA554698, PRJNA557339, PRJNA562050, PRJNA563077, PRJNA573690, PRJNA577465, PRJNA578299, PRJNA68459, and PRJNA84.

Links for the websites and software used in this study are as follows: Bactopia website and documentation, https://bactopia.github.io/; Github, https://www.github.com/bactopia/bactopia/; Zenodo Snapshot, https://doi.org/10.5281/zenodo.3926909; Bioconda, https://bioconda.github.io/recipes/bactopia/README.html; and the containers Docker, https://cloud.docker.com/u/bactopia/, and Singularity, https://cloud.sylabs.io/library/rpetit3/bactopia.

## SUPPLEMENTAL MATERIAL

Supplemental material is available online only.

**FIG S1**, PDF file, 0.1 MB.

**FIG S2**, PDF file, 0 MB.

**FIG S3**, PDF file, 0.02 MB.

**FIG S4**, PDF file, 0 MB.

**TABLE S1**, DOCX file, 0.02 MB.

**TABLE S2**, DOCX file, 0.02 MB.

**DATA SET S1**, TXT file, 2.4 MB.

**DATA SET S2**, TXT file, 0.02 MB.

**DATA SET S3**, PDF file, 0.1 MB.

## ACKNOWLEDGMENTS

We thank Torsten Seemann, Oliver Schwengers, Narciso Quijada, Michelle Su, Michelle Wright, Matt Plumb, Sean Wang, Ahmed Babiker, and Monica Farley for their helpful suggestions and feedback. We also acknowledge our gratitude to the many scientists and their funders who provided genome sequences to the public domain, to ENA and SRA for storing and organizing the data, and to the authors of the open source software tools and data sets used in this work.

Support for this project came from an Emory Public Health Bioinformatics Fellowship funded by the CDC Emerging Infections Program (U50CK000485) PPHF/ACA: Enhancing Epidemiology and Laboratory Capacity.

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
