## [Reviewer comments · mSystems]

Bactopia: a flexible pipeline for complete analysis of bacterial genomes

Robert Petit and Timothy Read

Corresponding Author(s): Timothy Read, Emory University School of Medicine

Review Timeline:

Submission Date:	April 23, 2020
Editorial Decision:	May 30, 2020
Revision Received:	July 7, 2020
Accepted:	July 15, 2020

Editor: Nicola Segata

Reviewer(s): The reviewers have opted to remain anonymous.

Transaction Report:

DOI: <https://doi.org/10.1128/mSystems.00190-20>

May 30, 2020

Dr. Timothy D Read
Emory University School of Medicine
Atlanta

Re: mSystems00190-20 (Bactopia: a flexible pipeline for complete analysis of bacterial genomes)

Dear Dr. Timothy D Read: thanks for your submission. I'm happy to tell you that the reviewers found your paper of interest (see below). They however raised several points that should be addressed in a revised manuscript before taking a final decision.

Below you will find the comments of the reviewers.

To submit your modified manuscript, log onto the eJP submission site at <https://msystems.msubmit.net/cgi-bin/main.plex>. If you cannot remember your password, click the "Can't remember your password?" link and follow the instructions on the screen. Go to Author Tasks and click the appropriate manuscript title to begin the resubmission process. The information that you entered when you first submitted the paper will be displayed. Please update the information as necessary. Provide (1) point-by-point responses to the issues raised by the reviewers as file type "Response to Reviewers," not in your cover letter, and (2) a PDF file that indicates the changes from the original submission (by highlighting or underlining the changes) as file type "Marked Up Manuscript - For Review Only."

Due to the SARS-CoV-2 pandemic, our typical 60 day deadline for revisions will not be applied. I hope that you will be able to submit a revised manuscript soon, but want to reassure you that the journal will be flexible in terms of timing, particularly if experimental revisions are needed. When you are ready to resubmit, please know that our staff and Editors are working remotely and handling submissions without delay. If you do not wish to modify the manuscript and prefer to submit it to another journal, please notify me of your decision immediately so that the manuscript may be formally withdrawn from consideration by mSystems.

To avoid unnecessary delay in publication should your modified manuscript be accepted, it is important that all elements you upload meet the technical requirements for production. I strongly recommend that you check your digital images using the Rapid Inspector tool at <http://rapidinspector.cadmus.com/RapidInspector/zmw/>.

Sincerely,

Nicola Segata

Editor, mSystems

Journals Department
Reviewer comments:

Reviewer #1 (Comments for the Author):

In this manuscript, the authors present a comprehensive pipeline for the automation of computational analysis tasks for bacterial genomes. Their pipeline is written in Nextflow, which is well known for its computational flexibility. The authors should be commended on the documentation available for their tool, as it is very thorough. Ultimately a pipeline of this flexibility is needed in the bacterial genomics community, especially to assist those groups with less computational experience. However, I do have some comments below that I feel will improve the manuscript.

1. Bactopia datasets section - I think it's great that the pipeline is able to automate downloading public datasets. However, there is no discussion about how easy/difficult it is to alter these databases, or clean them. Public datasets often contain erroneous content, or in the case of RefSeq, sequences that are incorrectly assigned. Ensuring that your datasets are as complete and accurate as possible greatly influences your downstream analysis. Some information on the format of these databases and how to edit them (or links to the downstream tool that gives information on database format?) would be useful. This information could go in the documentation rather than the manuscript
2. Line 112 - Sometimes there is more than one MLST scheme for an organism, and the user might want one of them, or both. How does Bactopia deal with this? If not mentioned in the manuscript directly, I think this should be covered in the documentation.
3. Line 114 - When downloading RefSeq genomes, is there any way to control this to prevent large downloads? Eg if you are working with Salmonella or Escherichia, there are thousands and thousands of public genomes, which would take a long time to download and take up a large amount of disk space. Is there a way to use the mash sketch to download a representative sample of genomes for a given taxID? Or to provide a list of public genomes the user would like to include (perhaps they have pre-filtered by ST)
4. Line 148 - Can you provide run or biosample accessions instead of experiment accessions? Many studies provide only run or biosample accessions in their supplementary material. Secondly, how does the pipeline handle multiple Illumina runs for the same experiment accession? Does it just download the most recent?

5. Line 199 - is this supposed to be sequence contaminant, not containment? If you do mean screening for sequence contaminants, how are contaminants detected, and what are the thresholds? This element of QC seems pretty key to me.
6. Line 315 - What was the rationale for the default cutoffs here for read and assembly quality? Do you need to meet all these cutoffs to be grouped at a particular level, or only some of them?
7. Line 324 - What is the difference between Exclude and QC failure? How are these decisions reached?
8. In general, I find the example bactopia commands provided in the text to be very helpful. I'm aware that the explanation for each flag will be in the pipeline documentation, but I think it would be useful to the reader if the flags used in the commands provided in the manuscript was explained, the first time they were provided. Eg it wasn't clear to me until I read the documentation that "include_genus" specified to the pipeline that the genomes should be downloaded by RefSeq
9. Figure 2 appears to be very low resolution in the pdf version provided to me, it was difficult to see the branches

Reviewer #2 (Comments for the Author):

The authors present a useful and comprehensive pipeline, named Bactopia, for the analysis of large collections of bacterial genomes. They have incorporated the most recent state-of-the-art tools and have released their package as a nextflow workflow to ensure its reproducible and portable use. Although other alternatives exist to perform similar set of analysis, the authors have done a good job highlighting how their approach differs from existing pipelines.

However, I was a bit disappointed that the authors decided to solely focus their workflow on the analysis of Illumina-derived sequences. Long-read sequencing, especially using the MinION platform is becoming more and more frequently used. Although the authors admit that this is a plan for future releases, I would like to stress the importance of long-read compatibility for the future viability of their pipeline and its wider use.

Major:

1) I would suggest the authors to implement quality-control (QC), assembly and polishing steps specific for long-read sequencing data to some extent in their current release (preferably for both nanopore and PacBio data). Ideally, there would also be an option to perform hybrid analysis if the user has both Illumina and long-read data. There are a number of tools that have been developed for this purpose. I provide below some suggestions:

Assembly: 'Canu' (PMID: 28298431) for long-read data only, or 'Unicycler' (PMID: 28594827) for hybrid datasets (Illumina and long-read data).

Polishing: 'Racon' (PMID: 28100585) for initial polishing, followed by 'Nanopolish' (<https://github.com/jts/nanopolish>) for Oxford nanopore data, 'Arrow' (<https://github.com/PacificBiosciences/GenomicConsensus>) for PacBio or 'Pilon' (PMID: 25409509), which is already implemented in their Illumina pipeline.

2) One crucial aspect missing from the current pipeline is a quality control of the resulting genome assemblies. Not only are assembler algorithms error-prone, but there is a risk of lab/reagent contamination that might introduce foreign sequences into the assembly. Implementing both an assembly QC step with QUAST, as well as completeness/contamination estimates with CheckM (to

filter assemblies with >5% contamination) is imperative to ensure the results of downstream analysis are robust. In addition, when processing assemblies derived from long-read data, an additional control could be performed to quantify the number of pseudogenes/truncated proteins in their assembly that could indicate a high sequence error rate. By comparing the observed number of pseudogenes with a general distribution expected for reference genomes of the same species, this would inform about whether a substantial number of artefactual frameshifts/indels are still present.

Moderate:

3) To increase the general usefulness of the pipeline, it would be a good idea to allow users to directly provide a set of genome assemblies (FASTA) as input instead of FASTQs. This means the pipeline would skip the assembly step and move straight into the QC steps (QUAST, CheckM) followed by annotation, pan-genome analysis, etc. Many users will probably just want to use already existing assemblies, so having this option would add great value to the pipeline.

Minor:

4) Although less critical than the previous points, it would be more informative to provide within their Bactopia pipeline a more thorough functional characterization of the predicted proteins. This could be performed with a tool such as eggNOG-mapper (PMID: 28460117).

5) The schematic provided in Fig. 1 is not very informative in my opinion. It would be more useful to provide a diagram of the steps and tools used in each process, so users understand how the different tools are linked together. This could be provided as a simplified diagram in Fig. 1, but also in more detail as a Supplementary Figure.

Dear Dr. Segata,

We would like to thank you and the reviewers for reviewing our manuscript: *Bactopia: a flexible pipeline for complete analysis of bacterial genomes*. The feedback was very useful for improving Bactopia.

There were relatively few changes recommended that directly affected the manuscript itself. We did reorganize the manuscript to comply with *MSystems* in-house style (ie using Results, Material and Methods, Importance, Discussion sections) but this involved moving sections of text rather than editing. Most of the reviewer recommendations were for improvements and additions to the Bactopia software pipeline, although these changes did not affect the analysis presented in the manuscript. The recommendations were the addition of alternative inputs, assembly assessment, and more Bactopia Tools. In our detailed responses describing the changes below we have included links to GitHub commits and links to updated software documentation.

After thorough testing of the changes, we released a new version of Bactopia, version 1.4.0 (<https://github.com/bactopia/bactopia/releases/tag/v1.4.0>). The new version is available from Bioconda, Docker, and Singularity.

Again, thank you and the reviewers for your time and effort.

Robert Petit and Tim Read

Reviewer #1 (Comments for the Author):

In this manuscript, the authors present a comprehensive pipeline for the automation of computational analysis tasks for bacterial genomes. Their pipeline is written in Nextflow, which is well known for its computational flexibility. The authors should be commended on the documentation available for their tool, as it is very thorough. Ultimately a pipeline of this flexibility is needed in the bacterial genomics community, especially to assist those groups with less computational experience. However, I do have some comments below that I feel will improve the manuscript.

1. Bactopia datasets section - I think it's great that the pipeline is able to automate downloading public datasets. However, there is no discussion about how easy/difficult it is to alter these databases, or clean them. Public datasets often contain erroneous content, or in the case of RefSeq, sequences that are incorrectly assigned. Ensuring that your datasets are as complete and accurate as possible greatly influences your downstream analysis. Some information on the format of these databases and how to edit them (or links to the downstream tool that gives

information on database format?) would be useful. This information could go in the documentation rather than the manuscript

We have improved the documentation with respect to adding species-specific custom protein sets, mash sketches and custom MLST schemas. The process for adding user curated protein sets and minmer sketches was also improved in the #3 comment below.

The documentation already contained information on adding sequences for blast and mapping and genomes for variant calling.

GitHub Commits:

<https://github.com/bactopia/bactopia/commit/7866fd691e38f57c8af011bdd50bbdb2112ea299>

Documentation:

<https://bactopia.github.io/datasets/#datasets-folder-overview>

2. Line 112 - Sometimes there is more than one MLST scheme for an organism, and the user might want one of them, or both. How does Bactopia deal with this? If not mentioned in the manuscript directly, I think this should be covered in the documentation.

We have added support for multiple MLST schemas. If an organism has multiple MLST schemas available from pubMLST.org (~7 organisms), each schema is downloaded and set up for the user. Bactopia will then call MLST against each available schema (e.g. E. coli's Achtman and Pasteur schemas). We also included links in the documentation for users to follow in the case they would like to create their own schema.

We have highlighted this change in the documentation.

GitHub Commits:

<https://github.com/bactopia/bactopia/commit/4f236b6eff462ed13139298b5eaa0a00e3c382aa>

<https://github.com/bactopia/bactopia/commit/8e69019d8e36efe7dcb7b9c9cd8b0567c954f187>

<https://github.com/bactopia/bactopia/commit/83903dc95fd726ed740aa544d46c72570067b2a2>

Documentation Links:

<https://bactopia.github.io/datasets/#mlst>

<https://bactopia.github.io/datasets/#mlst>

3. Line 114 - When downloading RefSeq genomes, is there any way to control this to prevent large downloads? Eg if you are working with Salmonella or Escherichia, there are thousands and thousands of public genomes, which would take a long time to download and take up a large amount of disk space. Is there a way to use the mash sketch to download a representative

sample of genomes for a given taxID? Or to provide a list of public genomes the user would like to include (perhaps they have pre-filtered by ST)

We have added the ability to limit the number of genomes downloaded during the dataset creation step as well as within certain Bactopia Tools. The user can now use the parameter '--limit' to specify the maximum number of genomes to be downloaded. In cases where there are more available completed genomes than the limit, a random subset of genomes is selected.

We have also added the parameter '--accessions' which allows the user to specify a list of RefSeq Assembly (e.g. GCF_*) accessions to download. This allows advanced users to create a subset of samples (as suggested above) that better fit their study.

We have included updates in the documentation to highlight these changes.

GitHub Commits:

<https://github.com/bactopia/bactopia/commit/0411da5c42f623e00cdac8b0992966dd100db32b>
<https://github.com/bactopia/bactopia/commit/c7fab1eec89c26f599a043354ed40c7cc883766e>
<https://github.com/bactopia/bactopia/commit/82ec3fd83fc1e1b58e29f8a60d1bd936797d698a>
<https://github.com/bactopia/bactopia/commit/a686d6e3939c6d833b93a81fdcd84e1bf41053e6>

Documentation Links:

<https://bactopia.github.io/datasets/#-limit>
<https://bactopia.github.io/datasets/#-accessions>

4. Line 148 - Can you provide run or biosample accessions instead of experiment accessions? Many studies provide only run or biosample accessions in their supplementary material. Secondly, how does the pipeline handle multiple Illumina runs for the same experiment accession? Does it just download the most recent?

Due to the accession hierarchy used by SRA/ENA, the Experiment accession is the only truly unique accession associated to raw sequences. BioSamples can have multiple Experiment accessions associated with it (Example: SAMN00792132, https://www.ncbi.nlm.nih.gov/sra?LinkName=biosample_sra&from_uid=792132) which can lead to confusion as whether the Experiments and associated sequences are the same or not. Due to this non-unique nature of BioSamples, we chose to work with Experiment accessions.

Although to help users with BioSamples, we added the ability to search BioSample and Run accessions to the 'bactopia search' tool. For a given BioSample, a list of all associated Experiment accessions will be produced. Users can then use this list of Experiment accessions with Bactopia.

In cases where an Experiment may have multiple runs, we have chosen to merge the Runs into a single representative set.

GitHub Commits:

<https://github.com/bactopia/bactopia/commit/9f31626abd7796a2a7f450a244d7b455cdf87f50>
<https://github.com/bactopia/bactopia/commit/83eca073bf97baf92135dfdebe6355136e0b1496>

Documentation Links:

<https://bactopia.github.io/usage-basic/#generating-accession-list>

5. Line 199 - is this supposed to be sequence contaminant, not containment? If you do mean screening for sequence contaminants, how are contaminants detected, and what are the thresholds? This element of QC seems pretty key to me.

This is sequence containment estimation using Mash Screen. We have reworded this sentence from “sequence containment screening against RefSeq with mash screen” to “sequence containment estimation against RefSeq with mash screen” (LINES 190-192).

Bactopia itself only removes PhiX and Illumina primer contaminants, it does not identify and remove other forms of contaminants. Instead we have implemented ‘mash screen’ and ‘sourmash lca gather’ for the user to review the results in case there is suspicion their data is not what they thought it was.

6. Line 315 - What was the rationale for the default cutoffs here for read and assembly quality? Do you need to meet all these cutoffs to be grouped at a particular level, or only some of them?

The default cutoffs were selected based on our previous work with Staphopia (<https://staphopia.emory.edu>) and personal preference. We have added text to note that the defaults are based on the Staphopia Analysis Pipeline (LINES 305-306). We also added the text on LINES 304-305 to highlight each cutoff had to be met to be placed in a group.

7. Line 324 - What is the difference between Exclude and QC failure? How are these decisions reached?

We renamed ‘QC Failure’ to ‘Unprocessed’ to more explicitly state the difference between Exclude (processed but failed cutoffs) and Unprocessed (not processed by Bactopia due technical limitations, e.g. low read count or poor genome estimate).

The idea is that the “Exclude” set met the minimum requirements for Bactopia processing but for the purposes of the Lactobacillus analysis presented here, we wanted only the best quality assemblies and hence excluded these projects.

The cutoffs for ‘Unprocessed’ were based on the range of expected Lactobacillus genome sizes stated at LINES 280-281.

8. In general, I find the example bactopia commands provided in the text to be very helpful. I'm aware that the explanation for each flag will be in the pipeline documentation, but I think it would be useful to the reader if the flags used in the commands provided in the manuscript was explained, the first time they were provided. Eg it wasn't clear to me until I read the documentation that "include_genus" specified to the pipeline that the genomes should be downloaded by RefSeq

After reviewing the command in question ‘--include_genus’ turned out to be redundant (--species ‘Lactobacillus’ was already genus only) and was removed from the text. But in the paragraphs following the commands we have included the parameters in the description.

See LINES 268, 276-278, 280-281, 347, 354-357, 403, 405, 408

9. Figure 2 appears to be very low resolution in the pdf version provided to me, it was difficult to see the branches

We have uploaded high quality images.

Reviewer #2 (Comments for the Author):

The authors present a useful and comprehensive pipeline, named Bactopia, for the analysis of large collections of bacterial genomes. They have incorporated the most recent state-of-the-art tools and have released their package as a nextflow workflow to ensure its reproducible and portable use. Although other alternatives exist to perform similar set of analysis, the authors have done a good job highlighting how their approach differs from existing pipelines.

However, I was a bit disappointed that the authors decided to solely focus their workflow on the analysis of Illumina-derived sequences. Long-read sequencing, especially using the MinION platform is becoming more and more frequently used. Although the authors admit that this is a plan for future releases, I would like to stress the importance of long-read compatibility for the future viability of their pipeline and its wider use.

Major:

1) I would suggest the authors to implement quality-control (QC), assembly and polishing steps specific for long-read sequencing data to some extent in their current release (preferably for both nanopore and PacBio data). Ideally, there would also be an option to perform hybrid analysis if the user has both Illumina and long-read data. There are a number of tools that have been developed for this purpose. I provide below some suggestions:

Assembly: 'Canu' (PMID: 28298431) for long-read data only, or 'Unicycler' (PMID: 28594827) for hybrid datasets (Illumina and long-read data).

Polishing: 'Racon' (PMID: 28100585) for initial polishing, followed by 'Nanopolish' (<https://github.com/jts/nanopolish>) for Oxford nanopore data, 'Arrow' (<https://github.com/PacificBiosciences/GenomicConsensus>) for PacBio or 'Pilon' (PMID: 25409509), which is already implemented in their Illumina pipeline.

We have added hybrid assembly using Unicycler. Users must provide paired-end Illumina reads along with the long reads. The Illumina reads are used through the pipeline (mapping, variant calling, etc..) and the long reads are only used at the assembly step.

By default Racon and Pilon are executed in the Unicycler pipeline. Users have the option to turn that off if they want to.

This hybrid assembly is an experimental feature limited to local data at this time. Integrating hybrid and de novo assemblies into the same pipeline as de novo Illumina assemblies is a long-term goal of the Bactopia project but will take a considerable amount of development and testing that we believe that is outside the scope of the current publication.

GitHub Commits:

<https://github.com/bactopia/bactopia/commit/18311dd324648d3d1527bbf7d39bdd29a8f25336>

<https://github.com/bactopia/bactopia/commit/5418c71333b37aed6e5ec7f4a0c3d468954e4425>

<https://github.com/bactopia/bactopia/commit/6132bd8d0d565d0f2f582cd68158b66efaa534c8>

Documentation Links:

<https://bactopia.github.io/usage-basic/#single-sample>

<https://bactopia.github.io/output-overview/#hybrid>

<https://bactopia.github.io/usage-complete/#required>

2) One crucial aspect missing from the current pipeline is a quality control of the resulting genome assemblies. Not only are assembler algorithms error-prone, but there is a risk of lab/reagent contamination that might introduce foreign sequences into the assembly. Implementing both an assembly QC step with QUAST, as well as completeness/contamination estimates with CheckM (to filter assemblies with >5% contamination) is imperative to ensure the results of downstream analysis are robust. In addition, when processing assemblies derived from long-read data, an additional control could be performed to quantify the number of pseudogenes/truncated proteins in their assembly that could indicate a high sequence error rate. By comparing the observed number of pseudogenes with a general distribution expected for reference genomes of the same species, this would inform about whether a substantial number of artefactual frameshifts/indels are still present.

We have added QUAST and CheckM to the main Bactopia Analysis Pipeline. By default, CheckM uses the reduced tree due to the memory requirements versus the full tree (<16GB vs ~40GB). There is an option for users to use the full tree if they would like to.

GitHub Commits:

<https://github.com/bactopia/bactopia/commit/24c4b309b4c10fcba1da635f7c5138e2235d1df3>

<https://github.com/bactopia/bactopia/commit/4697590c493916bbf3905e52c7d548e557f24fa7>

<https://github.com/bactopia/bactopia/commit/5418c71333b37aed6e5ec7f4a0c3d468954e4425>

<https://github.com/bactopia/bactopia/commit/6132bd8d0d565d0f2f582cd68158b66efaa534c8>

Documentation Links:

<https://bactopia.github.io/workflow-overview/#assembly-quality-assessment>

<https://bactopia.github.io/output-overview/#quality-reports>

<https://bactopia.github.io/usage-complete/#assembly-quality-control-parameters>

Moderate:

3) To increase the general usefulness of the pipeline, it would be a good idea to allow users to directly provide a set of genome assemblies (FASTA) as input instead of FASTQs. This means the pipeline would skip the assembly step and move straight into the QC steps (QUAST, CheckM) followed by annotation, pan-genome analysis, etc. Many users will probably just want to use already existing assemblies, so having this option would add great value to the pipeline.

We have added support for assemblies as inputs. Users can provide their own assembly, or download assemblies from NCBI Assembly using GCA* or GCF* accessions. If an assembly is used, 2x250bp error-free Illumina reads are simulated for the assembly. By default the assembly will not be reassembled at the assembly step, but if the user wants they can reassemble using the simulated reads.

We chose to simulate reads based on the contigs and reassemble de novo in order for all steps in the main Bactopia pipeline to be executed. This would allow users to mix samples with FASTQs and FASTAs, while maintaining consistent outputs necessary for downstream analysis using the Bactopia Tools (e.g. roary, phyloflash).

GitHub Commits:

<https://github.com/bactopia/bactopia/commit/21cd950ab7961d4ae895e4d7dc46fae571d7ef0e>

<https://github.com/bactopia/bactopia/commit/18311dd324648d3d1527bbf7d39bdd29a8f25336>

Documentation Links:

<https://bactopia.github.io/usage-basic/#single-sample>

<https://bactopia.github.io/workflow-overview/#gather-fastqs>

<https://bactopia.github.io/output-overview/#assembly>

<https://bactopia.github.io/usage-complete/#required>

Minor:

4) Although less critical than the previous points, it would be more informative to provide within their Bactopia pipeline a more thorough functional characterization of the predicted proteins. This could be performed with a tool such as eggNOG-mapper (PMID: 28460117).

We have added a Bactopia Tool for eggNOG-mapper. Due to eggNOG-mapper requiring a ~40GB database, we felt it was best to go the Bactopia Tool route instead of adding it to the main Bactopia analysis pipeline. This also allows the user to pick and choose with samples to run eggNOG-mapper on, instead of all.

GitHub Commits:

<https://github.com/bactopia/bactopia/commit/e1264d10fab851ebb262dd0d04aebded8e4fd931>

<https://github.com/bactopia/bactopia/commit/291abb98356c9ba0c3f1d818b82d898c36cba7f6>

Documentation Link:

<https://bactopia.github.io/bactopia-tools/eggnog/>

5) The schematic provided in Fig. 1 is not very informative in my opinion. It would be more useful to provide a diagram of the steps and tools used in each process, so users understand how the different tools are linked together. This could be provided as a simplified diagram in Fig. 1, but also in more detail as a Supplementary Figure.

We have renamed Figure 1 to Figure 1a and added Figure 1b, an alternate version of Supplementary Figure 1. Figure 1b is a more compact version of Supplementary Figure 1, except with the addition of significant bioinformatic tools used at each step.

July 15, 2020

Dr. Timothy D Read
Emory University School of Medicine
Atlanta

Re: mSystems00190-20R1 (Bactopia: a flexible pipeline for complete analysis of bacterial genomes)

Dear Dr. Timothy D Read:

Thanks for your revision and work on the method. The reviewers are both finding your manuscript very improved and are happy with the paper.

Your manuscript has been accepted, and I am forwarding it to the ASM Journals Department for publication. For your reference, ASM Journals' address is given below. Before it can be scheduled for publication, your manuscript will be checked by the mSystems senior production editor, Ellie Ghatineh, to make sure that all elements meet the technical requirements for publication. She will contact you if anything needs to be revised before copyediting and production can begin. Otherwise, you will be notified when your proofs are ready to be viewed.

Sincerely,

Nicola Segata
Editor, mSystems

Journals Department
Supplemental Material: Accept
Supplemental Material: Accept
Data S2: Accept
Supplemental Material: Accept
Supplemental Material: Accept
Data S3: Accept
Supplemental Material: Accept
Data S1: Accept
Supplemental Material: Accept